# *Klebsiella pneumoniae* clinical isolates with features of both multidrug-resistance and hypervirulence have unexpectedly low virulence

Travis J. Kochan [1,2] ✉, Sophia H. Nozick[2], Aliki Valdes[2], Sumitra D. Mitra[2], Bettina H. Cheung[2], Marine Lebrun-Corbin [2], Rachel L. Medernach[2,3], Madeleine B. Vessely [2], Jori O. Mills[2], Christopher M. R. Axline [2], Julia A. Nelson[2], Ethan M. VanGosen [2], Timothy J. Ward[2], Egon A. Ozer [3], David van Duin[4], Liang Chen [5], Barry N. Kreiswirth [5], S. Wesley Long [6], James M. Musser[6], Zackery P. Bulman [7], Richard G. Wunderink [8,9] & Alan R. Hauser [2,3]

*Klebsiella pneumoniae* has been classified into two types, classical *K. pneumoniae* (cKP) and hypervirulent *K. pneumoniae* (hvKP). cKP isolates are highly diverse and important causes of nosocomial infections; they include globally disseminated antibiotic-resistant clones. hvKP isolates are sensitive to most antibiotics but are highly virulent, causing community-acquired infections in healthy individuals. The virulence phenotype of hvKP is associated with pathogenicity loci responsible for siderophore and hypermucoid capsule production. Recently, convergent strains of *K. pneumoniae*, which possess features of both cKP and hvKP, have emerged and are cause of much concern. Here, we screen the genomes of 2,608 multidrug-resistant *K. pneumoniae* isolates from the United States and identify 47 convergent isolates. We perform phenotypic and genomic characterization of 12 representative isolates. These 12 convergent isolates contain a variety of antimicrobial resistance plasmids and virulence plasmids. Most convergent isolates contain aerobactin biosynthesis genes and produce more siderophores than cKP isolates but not more capsule. Unexpectedly, only 1 of the 12 tested convergent isolates has a level of virulence consistent with hvKP isolates in a murine pneumonia model. These findings suggest that additional studies should be performed to clarify whether convergent strains are indeed more virulent than cKP in mouse and human infections.

*Klebsiella pneumoniae* is a highly diverse human pathogen and a major cause of antimicrobial resistance-related deaths globally[1,2]. Bacteria of this species has been categorized into two broadly defined groups: classical *K. pneumoniae* (cKP) isolates, which cause nosocomial infections in compromised patients, and hypervirulent *K. pneumoniae* (hvKP) isolates, which cause community-acquired, disseminated infections in healthy, immunocompetent individuals[3–8].

cKP isolates are highly diverse, consisting of hundreds of sequence types (STs) and capsule loci types (KLs)[9]. They are generally of low virulence, which may account for their predominance in hospitalized and immunocompromised patients[10,11]. A clinically important subset are multidrug-resistant cKP (MDR-cKP) strains, which are resistant to clinically relevant antibiotics in 3 or more classes. These strains include high-risk clones, which are globally disseminated STs (e.g., ST258 or ST307). MDR-cKP strains frequently produce extended-spectrum beta-lactamases (ESBLs), including CTX-M-15 and SHV-12, which confer resistance to third-generation cephalosporins[9,12,13]. Some MDR-cKP isolates are resistant to carbapenems through plasmid-encoded carbapenemases, such as KPC and NDM[13,14].

hvKP isolates are usually susceptible to antibiotics but are exceptionally virulent[15]. The hypervirulent phenotype has been attributed to two types of factors: mucoid regulators and siderophores[16,17]. Mucoid regulator genes, *rmpADC* and *rmpA2*, enhance capsule production and tenacity, resulting in a hypermucoviscosity (hmv) colony morphology[18,19]. These capsules allow the bacterium to resist the host's innate immune response by inhibiting phagocytosis, opsonization, and complement-mediated killing[15]. hmv also limits DNA movement into and out of the bacterial cell, restricting recombination and horizontal gene transfer[20], potentially explaining the restricted range of hvKP sequence types. Siderophores are molecules secreted by bacteria to scavenge iron. *K. pneumoniae* isolates may produce up to four siderophores. Biosynthesis genes for enterobactin (*ent*) are present in nearly all *K. pneumoniae* strains, whereas biosynthesis genes for aerobactin (*iuc*) and salmochelin (*iro*) are frequently present in hvKP strains. Biosynthesis genes for yersiniabactin (*ybt*) are present in some hvKP and some cKP isolates; they are frequently associated with the genes encoding the genotoxin colibactin (*clb*) as part of integrated and conjugative elements such as ICE*Kp10*[21].

Accurately distinguishing between hvKP and cKP strains is challenging and the definitions of these groups remain controversial[22]. Although hvKP strains are hmv, cKP strains can also have this attribute due to mutations within the capsule locus (*wzc*)[23]. Some studies have defined hvKP as simply the presence of *iuc* genes[24,25]. Russo and colleagues systematically evaluated five genes (*rmpA*, *rmpA2*, *peg-344*, *iucA*, and *iroB*) and one phenotype (hmv) for defining hvKP in a sample of 175 hvKP or cKP isolates[11]. They found that presence of *iucA* or *iroB* yielded predictive accuracies of 96% and 97%, respectively, compared to the hmv phenotype, which yielded an accuracy of 90%. However, the clinical implications of these biomarkers across diverse strain backgrounds remain unclear.

Evidence suggests that mouse models of infection, which capture the overall virulence potential of *K. pneumoniae* isolates, are highly accurate in distinguishing hvKP from cKP[11]. In a mouse model of pneumonia, hvKP strains cause pre-lethal illness at extremely low doses (some strains <100 CFU), making them easily distinguishable from cKP strains, which usually require doses of $10^7$–$10^9$ CFU to cause a similar severity of illness[11]. For these reasons, mouse models are attractive as a gold standard for identifying hvKP strains[11,26].

Of great concern is the recent emergence of *K. pneumoniae* strains with features of both hvKP and MDR-cKP. These convergent strains have the potential to cause difficult-to-treat invasive and disseminated infections in healthy individuals. Several reports have documented the existence of typical hvKP strains that have acquired ESBL or carbapenemase genes[27–30]. For example, in 2019 Karlson and colleagues reported an ST23 isolate from the U.S. that carried a $bla_{KPC-2}$ plasmid, and in 2016 Cheong and colleagues reported two ST23 isolates in Singapore that carried a $bla_{CTX-M-15}$ plasmid[27,30]. Others have noted highly antibiotic-resistant cKP strains that have acquired virulence genes usually found in hvKP strains[28,29,31,32]. For example, Wyres and colleagues reported that 19 of 22 convergent bloodstream isolates in Southeastern Asia were MDR-cKP isolates that had acquired aerobactin

biosynthesis genes. Some reports suggest that convergent strains have enhanced virulence in neutrophil phagocytosis assays or animal models of infection and a few suggest that they cause more severe disease in people[33–36]. For example, Gu and colleagues described five patients infected with an ST11 carbapenem-resistant hvKP strain in China and noted that all five patients died[33]. However, data supporting a high degree of virulence associated with convergent strains are limited[31,32,35].

To address this gap in the literature, we screened three large U.S. collections of highly antibiotic-resistant cKP for convergent isolates. We carried out whole-genome sequence analysis and conducted phenotypic tests for established virulence traits of *K. pneumoniae*.

## Results
### Detection of convergent isolates among U.S. antibiotic-resistant *K. pneumoniae* clinical isolates

For the purposes of this study, we refer to convergent *K. pneumoniae* as strains that both are resistant to third-generation cephalosporins (e.g., ceftriaxone) and contain hvKP virulence genes. We define hvKP virulence genes as *iucA*, *iroB*, *rmpA*, or *rmpA2*. To identify convergent isolates, we screened the genomes of 2608 isolates from three large collections for at least one of these four hvKP virulence genes.

These collections were as follows: (1) isolates resistant to third-generation cephalosporins from Northwestern Memorial Hospital in Chicago (NMH collection, $n = 237$), (2) isolates resistant to carbapenems from 49 U.S. hospitals participating in the CRACKLE-2 study (CRACKLE-2 collection, $n = 884$) and, (3) isolates resistant to third-generation cephalosporins from the Houston-Methodist Hospital System (Houston collection, $n = 1486$)[10,37–39]. One additional isolate, DHQP1701672 from the U.S. Center for Disease Control and Prevention, was included because it is one of the few reported carbapenem-resistant hvKP isolates from the U.S.[27] Most of the 2608 isolates were from infections of the urinary tract ($n = 1266$, 48.5%), lung ($n = 588$, 22.5%), bloodstream ($n = 287$, 11.0%), and wounds ($n = 249$, 9.5%) (Table 1).

The 2608 *K. pneumoniae* isolates analyzed in our study exhibited a dominance of two multilocus sequence types (STs) endemic to the U.S., ST258 and ST307 (Figs. 1A and S1A). These STs were primarily capsule loci KL106/KL107 and KL102, respectively. The remaining isolates demonstrated high diversity, comprising 217 STs and 121 KLs, including high-risk clonal groups like ST14, ST15, and ST16 (Figs. 1A, S1A, S1B and Supplementary Data 1). Overall, the most common STs were ST258 ($n = 1060$, 40.6%), ST307 ($n = 576$, 22.1%), ST16 ($n = 96$,

**Table 1 | Clinical source of isolate by collection**

| Source | Northwestern n (%) | CRACKLE-2 n (%) | Houston n (%) | CDC n (%) | Total n (%) |
|---|---|---|---|---|---|
| Urine | 95 (40.1) | 377 (42.6) | 793 (53.4) | 1 (100) | 1266 (48.5) |
| Lung | 49 (20.7) | 215 (24.3) | 324 (21.8) | – | 588 (22.5) |
| Blood | 48 (20.3) | 120 (13.6) | 119 (8.0) | – | 287 (11.0) |
| Wound | 12 (5.1) | 109 (12.3) | 128 (8.6) | – | 249 (9.5) |
| Abscess | – | – | 27 (1.8) | – | 27 (1.0) |
| Bile | 5 (2.1) | – | 5 (0.3) | – | 10 (0.4) |
| Rectal | 4 (1.7) | – | – | – | 4 (0.2) |
| CSF | 1 (0.4) | – | 3 (0.2) | – | 4 (0.2) |
| Peritoneal fluid | 1 (0.4) | – | – | – | 1 (0.0) |
| Pleural fluid | 1 (0.4) | – | – | – | 1 (0.0) |
| Other | 21 (8.9) | 63 (7.1) | 87 (5.9) | – | 171 (6.6) |
| Total | 237 | 884 | 1486 | 1 | 2608 |

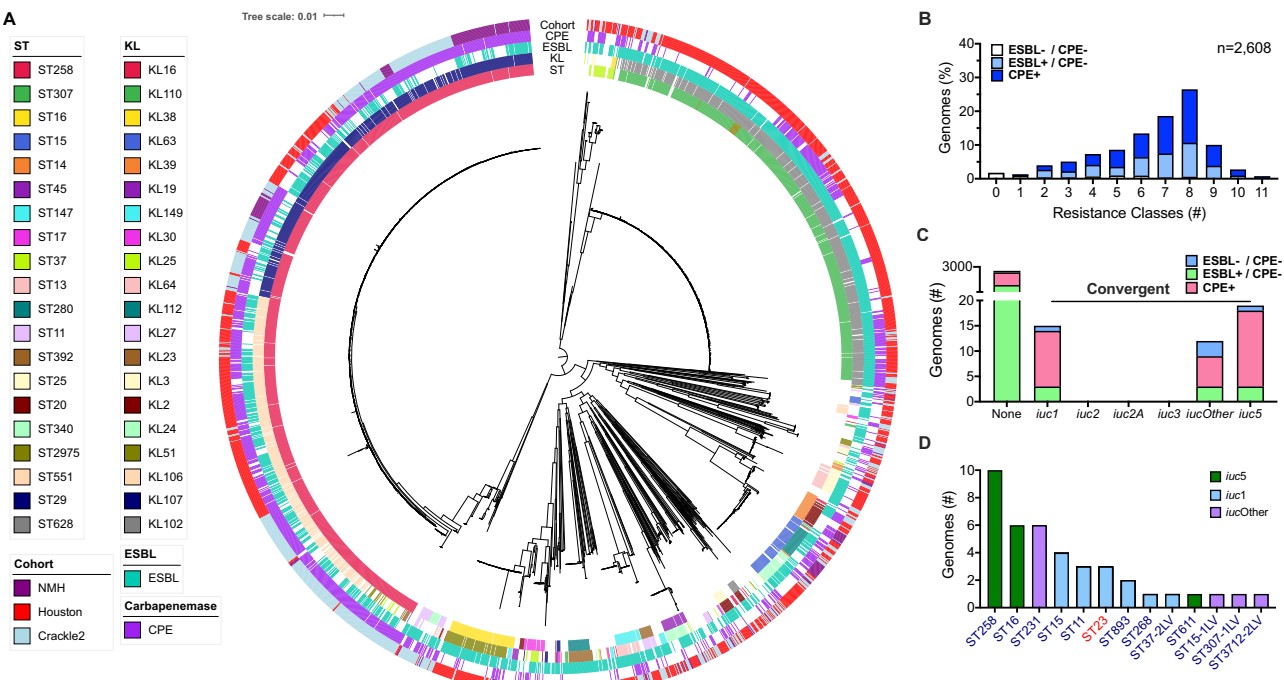

**Fig. 1 | Genomic features of antibiotic-resistant clinical isolates of *K. pneumoniae* from United States hospitals. A** A core genome phylogenetic tree of the 2608 isolates screened in this study. ST, KL type, isolate cohort, and presence of an ESBL or carbapenemase gene are indicated. **B** Distribution of the number of drug classes per genome for which antimicrobial resistance elements were detected. **C** The number of genomes that contained each of the different lineages of aerobactin (*iuc*). In (**B**) and (**C**), bars are colored based on the presence of an ESBL or carbapenemase producing enterobacteriaceae (CPE). **D** The distribution of convergent isolates by sequence type with the different lineages of *iuc*. cKP- and hvKP-associated STs are labeled in navy blue and red font, respectively. Source data are provided as a Source data file.

3.7%), ST15 (*n* = 90, 3.5%), and ST14 (*n* = 41, 1.6%) (Figs. 1A, S1A and Supplementary Data 1). The most common capsule loci were KL102 (*n* = 576, 22.1%), KL107 (*n* = 510, 19.6%), KL106 (*n* = 391, 15.0%), KL51 (*n* = 103, 3.9%), and KL24 (*n* = 71, 2.7%) (Figs. 1A, S1B and Supplementary Data 1).

Of the 2608 *K. pneumoniae* isolates examined, ~90% contained resistance determinants for 3 or more classes of antibiotics (Fig. 1B). While the majority contained an ESBL gene (*n* = 2132, 78.6%) or a carbapenemase gene (*n* = 1165, 43%), some isolates (*n* = 174, 6.7%) lacked these genes and were presumably resistant to third-generation cephalosporins or carbapenems by other mechanisms (Supplementary Data 1)[40,41].

Further screening identified 47 (1.8%) isolates with at least one hvKP virulence gene (*iucA*, *iroB*, *rmpA*, and *rmpA2*) and were deemed convergent: 25 (2.8%) of CRACKLE-2, 12 (0.8%) of Houston, and 9 (3.8%) of NMH isolates (Supplementary Data 1). Among these, 45 (95.7%) contained *iucA* with distinct lineages: 18 (40%) contained *iuc*5 (a lineage associated with *Escherichia coli* plasmids), 15 (33.3%) contained *iuc*1 (a hvKP-associated lineage), and 12 (26.7%) contained novel lineages of *iuc* (*iuc*Other) (Fig. 1C). None contained *iuc*2, a second hvKP-associated *iuc* lineage. Blast analysis of the 12 *iuc*Other lineages revealed they are slight variations of lineages found in *E. coli* plasmids. Of the 47 convergent isolates, 17 (36.2%) contained *iroB* and 14 (29.8%) contained either *rmpA* or *rmpA2* (Supplementary Data 1). Multilocus sequence typing of these isolates revealed that 44 (93.6%) were STs associated with cKP lineages, indicating that these highly antibiotic-resistant lineages had acquired virulence genes such as *iucA* or *iroB* (Fig. 1D)[42]. The most common of these STs were ST258 (*n* = 10) and ST16 (*n* = 6) (Fig. 1D). The remaining three isolates were ST23, a genotype commonly associated with hvKP. These ST23 lineages had acquired the *bla*KPC-2 carbapenemase gene (Supplementary Data 1). These findings confirm the presence of rare convergent *K. pneumoniae* isolates in the U.S.

## Phylogenetic analysis of convergent isolates

Twelve of the 47 convergent isolates were chosen for further analysis based on their representation of 9 distinct STs (Table 2 and Supplementary Data 2). Among these isolates, 11 were cKP that acquired hvKP-like virulence factors. The remaining isolate (DHQP1701672) was a typical hvKP sequence type (ST23) that acquired an antibiotic-resistance gene (*bla*KPC-2). To further examine how these 12 isolates evolved to become convergent, we compared their whole-genome sequences to other well-characterized isolates from three *K. pneumoniae* groups: typical hvKP (*n* = 11), MDR-cKP (*n* = 11), and relatively antibiotic-susceptible cKP (NON-MDR-cKP) (*n* = 6) (Fig. 2 and Supplementary Data 2). A core-genome phylogenetic tree was generated to examine the relationships between these isolates (Fig. 2). Convergent isolates clustered together with non-convergent isolates of the same ST or clonal group. For example, the convergent isolate ARLG-3346 clustered with other ST258 isolates, and KPN1409 clustered with other ST15 isolates (Fig. 2). In addition, the convergent isolate K014 is ST268, an ST not traditionally associated with hvKP; however, it did cluster with a confirmed ST268 hvKP isolate from NMH, KPN13 (Fig. 2)[42]. This indicates that the convergent isolates are not phylogenetically distinct from closely related non-convergent isolates and likely have emerged through acquisition of virulence or antimicrobial resistance genes or plasmids (Fig. 2).

## Plasmid analysis of convergent isolates

To evaluate plasmid content of the 12 convergent isolates, we performed long-read sequencing. For comparison, complete genomes were obtained, including plasmids, for all 11 of the reference hvKP isolates, all 11 of the MDR-cKP isolates, and all 6 of the NON-MDR-cKP isolates. Plasmid sequence similarity was visualized using a neighbor-joining tree based on Mash distance (Fig. 3). Several observations were apparent: (1) Many of the convergent isolates contained multiple plasmids. For example, KPN1402 and KPN482 each contained five

**Table 2 | Characteristics of convergent isolates chosen for further analysis**

| Strain | ST | K_locus | Ybt | Clb | Iuc | Iro | *rmpADC* | *rmpA2* | ESBL | Carbapenemase | hmv | Source |
|---|---|---|---|---|---|---|---|---|---|---|---|---|
| ARLG-3346 | ST258 | KL107 | *ybt 17* | *clb 3* | *iuc 5* | *iroOther* | – | – | – | *bla*KPC-3 | – | Lung |
| ARLG-4365 | ST11 | KL64 | *ybt 9* | – | *iuc 1* | *iro 1* | – | – | *bla*CTX-M-65; *bla*SHV-12 | *bla*KPC-2 | – | Urine |
| ARLG-4803 | ST231 | KL51 | *ybt 14* | – | *iuc 5* | – | – | – | *bla*CTX-M-15 | – | – | Blood |
| CRE-015 | ST16 | KL51 | – | – | *iuc 5* | – | – | – | *bla*TEM-116 | *bla*KPC-3 | – | Lung |
| DHQP1701672 | ST23 | KL1 | *ybt 1* | *clb 2* | *iuc 1* | *iro 1* | *rmpADC* | *rmpA2* | – | *bla*KPC-2 | – | Urine |
| K014 | ST268 | KL20 | *ybt 0* | *clb 3* | *iuc 1* | *iro 1* | *rmpADC* | *rmpA2* | *bla*SHV-12 | – | + | Blood |
| KPN1402 | ST15 | KL112 | *ybt 16* | – | *iuc 1* | – | – | *rmpA2* | *bla*CTX-M-15 | *bla*OXA-48 | – | Bile |
| KPN1409 | ST15 | KL112 | *ybt 16* | – | *iuc 1* | – | – | *rmpA2* | *bla*CTX-M-15 | *bla*OXA-48 | – | Blood |
| KPN226 | ST16 | KL51 | *ybt 2* | – | – | *iro 3* | *rmpADC* | – | *bla*CTX-M-15 | – | – | Urine |
| KPN415 | ST611 | KL71 | – | – | *iuc 5* | – | – | – | *bla*CTX-M-15 | – | – | Urine |
| KPN482 | ST15 | KL24 | – | – | *iuc 1* | – | – | *rmpA2* | *bla*SHV-2a | – | – | Lung |
| OC1092 | ST16 | KL51 | *ybt 17* | *clb 3* | *iuc 5* | – | – | – | – | *bla*KPC-3 | – | Blood |

*ST* sequence type, *K_locus* capsule locus, *ybt* yersiniabactin, *Clb* colibactin, *Iuc* Aerobactin, *hmv* hypermucoviscous by string test.

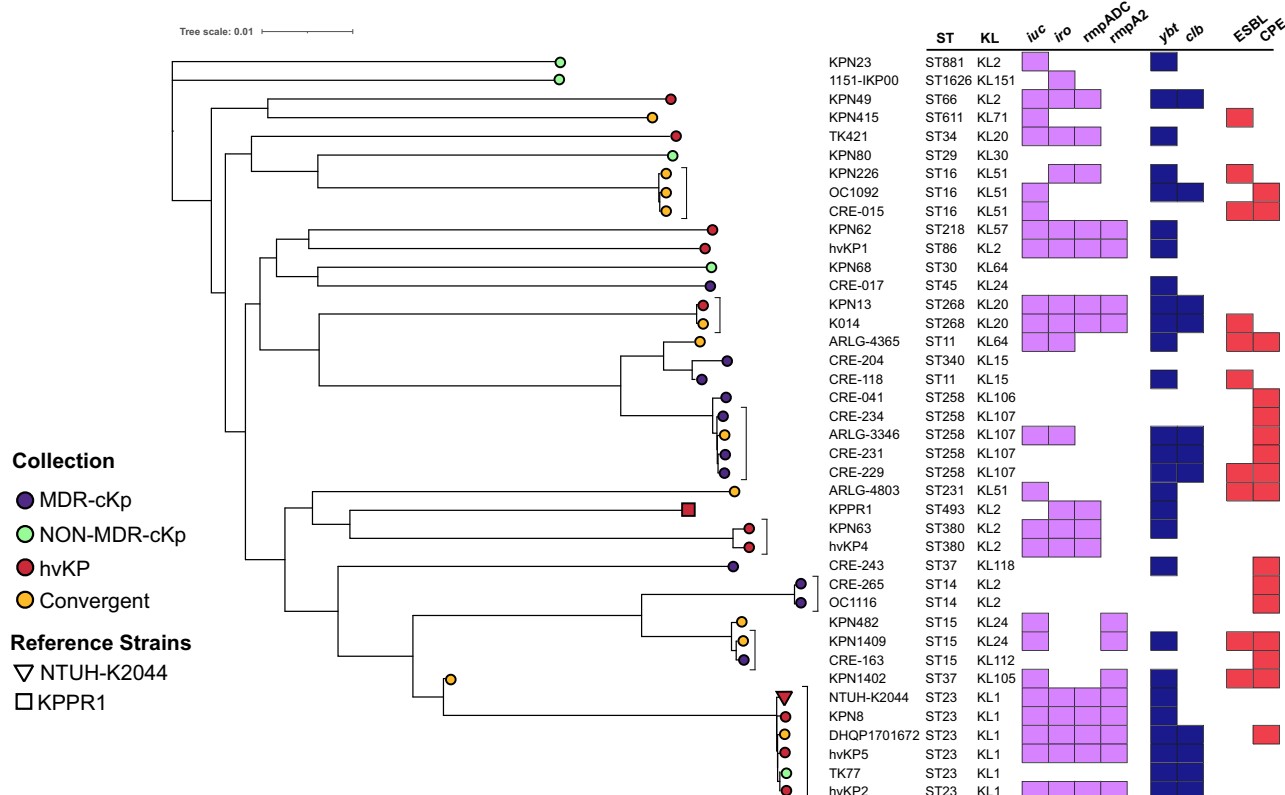

**Fig. 2 | Phylogenetic analysis of convergent, MDR-cKP, NON-MDR-cKp, and hvKP isolates.** A core genome phylogenetic tree of convergent isolates along with representative isolates of MDR-cKP, NON-MDR-cKp, and hvKP groups was generated. The presence of ST, KL type, virulence genes, and ESBL or carbapenemase genes are indicated. Two well-characterized hvKP reference strains are included: NTUH-K2044 and KPPR1. *iuc* = aerobactin biosynthesis genes, *iro* = salmochelin biosynthesis genes, *ybt* = yersiniabactin biosynthesis loci, *clb* = colibactin biosynthesis loci, *rmpADC* = mucoid regulator operon, *rmpA2* = mucoid regulator 2 gene, ESBL = extended-spectrum beta-lactamase gene, CPE = carbapenemase producing Enterobacteriaceae.

plasmids. (2) Many plasmids had multiple origins of replication, suggesting that they were formed by plasmid fusion. (3) Plasmids containing virulence genes clustered into three major groups. (4) Antibiotic resistance genes, encoding carbapenemases, ESBLs, and aminoglycoside modifying enzymes, were distributed broadly across plasmids, indicating their mobility and ability to integrate into different plasmids. (5) Although most plasmids containing the *bla*KPC gene fell into two distinct groups, other examples of plasmids with this gene were also observed (pCRE-265_3, pOC1116_4, pARLG-4365_2, and pCRE-015_3).

Eleven convergent isolates harbored plasmids containing virulence genes (Supplementary Data 3). Six convergent isolates contained plasmids with segments and origins of replication (IncFIB and IncHI1B) highly similar to pK2044, a well-characterized hvKP virulence plasmid (Figs. 3, 4A and Supplementary Data 1). Three (DHQP1701672, ARLG-4365, and K014) contained plasmids with >80% of the sequence content of pK2044. All six contained regions that aligned to both *rmpA2* and the *iuc* locus, but the *rmpA2* genes contained frameshift mutations resulting in premature stop codons. Three of these six isolates also contained *iro* loci and two contained *rmpADC* (Fig. 4A, Supplementary

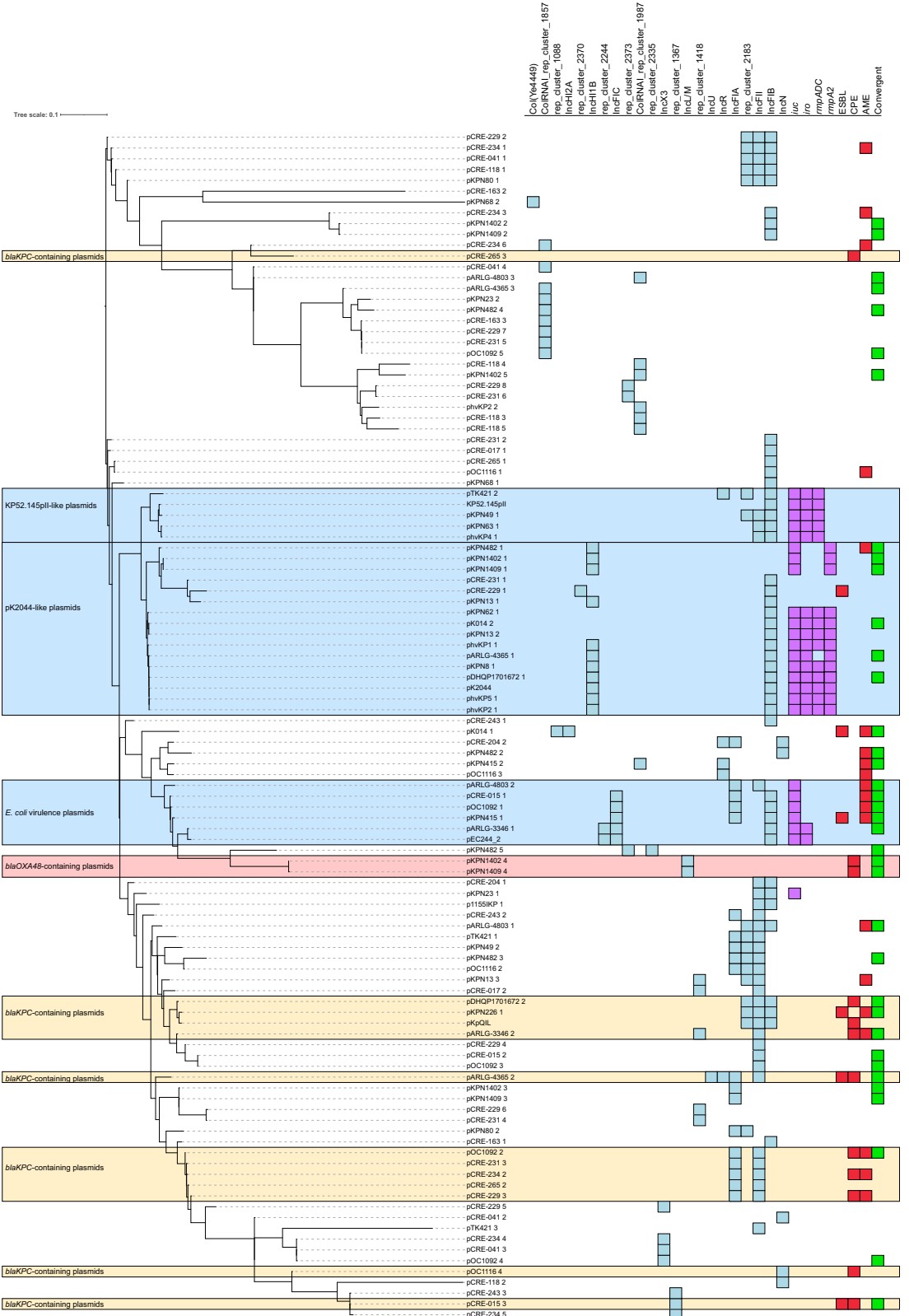

**Fig. 3 | Plasmid sequence analysis of convergent, MDR-cKP, hvKP, and NON-MDR-cKP isolates.** A neighbor-joining tree based on mash distance was constructed with all assembled plasmids from MDR-cKP, hvKP, and NON-MDR-cKP (Supplementary Data 3). Plasmid replicons, virulence genes, and classes of AMR genes are indicated. Plasmids from convergent isolates are indicated with green.

*iuc* = aerobactin biosynthesis genes, *iro* = salmochelin biosynthesis genes, *rmpADC* = mucoid regulator operon, *rmpA2* = mucoid regulator 2 gene, ESBL = extended-spectrum beta-lactamase gene, CPE = carbapenemase producing enterobacteriaceae, AME = aminoglycoside modifying enzyme gene.

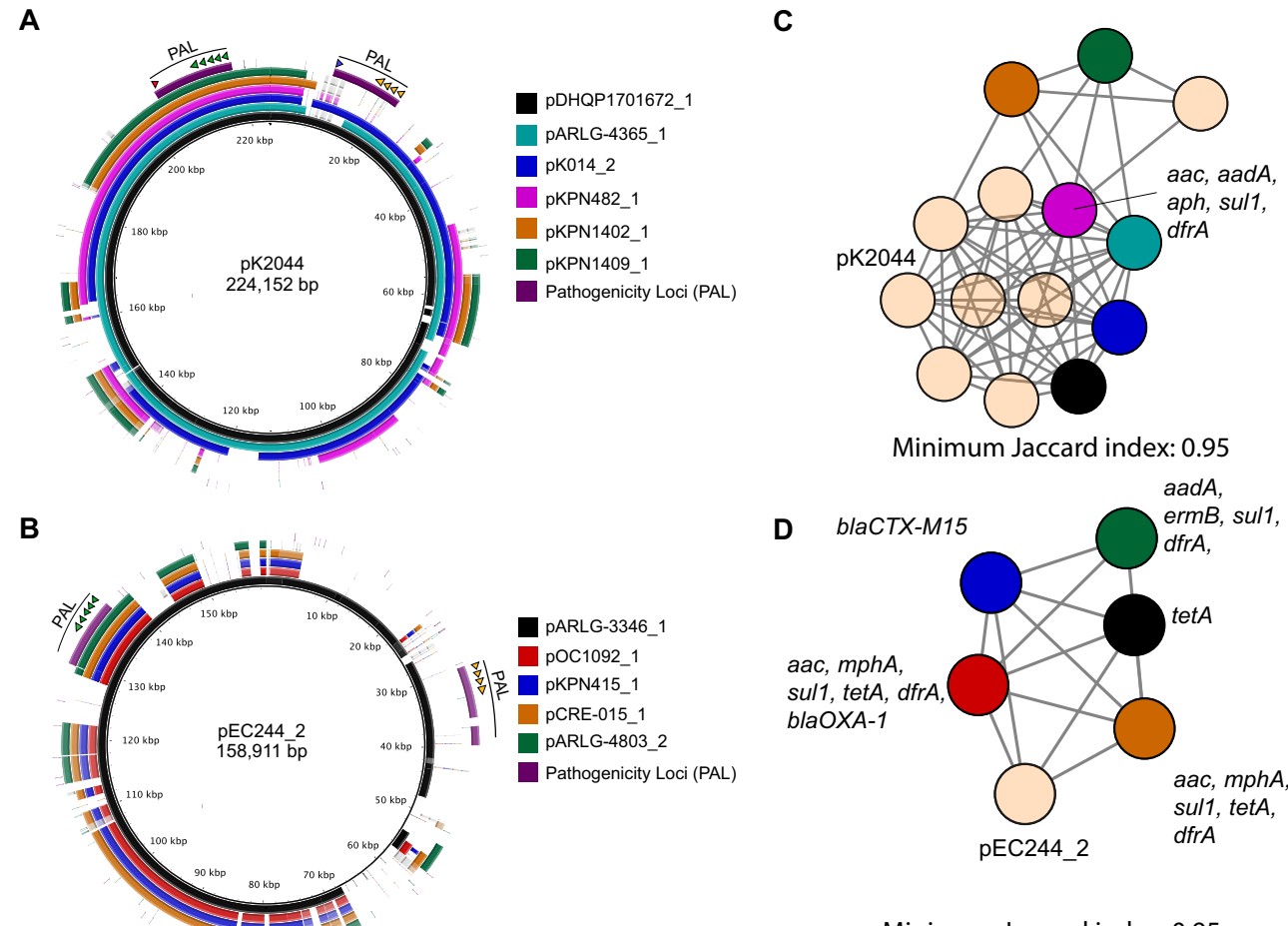

**Fig. 4 | Virulence plasmids identified in 12 representative convergent isolates.**
Plasmids from convergent isolates were aligned to **A** pK2044, a well-characterized hvKP virulence plasmid, or **B** pEC244_2, an *E. coli* plasmid. Alignments were made using BRIG with a sequence identity threshold of 85%. Aerobactin biosynthesis genes are indicated with green arrows, salmochelin with orange arrows, *rmpA* with a blue arrow, and *rmpA2* with a red arrow. De novo clustering of **C** pK2044-related plasmids and **D** pEC244_2-related plasmids was performed using Mash with a minimum Jaccard index of 0.95. Reference plasmids pK2044 and pEC244_2 were included in the analyses and are indicated. Individual nodes represent an assembled plasmid. Node colors in (**C**) match plasmid colors in (**A**), and node colors in (**D**) match plasmid colors in (**B**). Tan colored nodes represent plasmids from non-convergent isolates that had Jaccard index of 0.95 or higher with either pK2044 in (**C**) or pEC244_2 in (**D**). Networks were graphed using Cytoscape. In panels **C** and **D**, antibiotic-resistance genes present on plasmids are listed.

Data 3). These six plasmids grouped closely with other pK2044-like hvKP plasmids based on genomic similarity (Jaccard Index >0.95) (Fig. 4C). Five other convergent isolates harbored plasmid sequences and replicons (IncFIC, IncFIB, and rep_cluster_2244) similar to portions of pEC244_2, an *E. coli* virulence plasmid containing *iuc* and *iro* loci (Fig. 4B, Supplementary Data 1, and Supplementary Data 3). One plasmid (pARLG-3346_1) contained both *iuc* and *iro* loci while four contained only the *iuc* locus (Fig. 4B and Supplementary Data 1). Based on genomic similarity, these five plasmids grouped together (Jaccard Index >0.95); each carried at least one antimicrobial resistance gene (Fig. 4D). One of these plasmids, pKPN415_1 contained a $bla_{CTX-M-15}$ ESBL gene. The twelfth convergent isolate, KPN226, contained a chromosomal copy of ICE*Kp1*, which contained the *iro* locus and the *rmpADC* operon (Table 2 and Supplementary Data 1). These findings highlight the diverse plasmids in convergent isolates with various hvKP-associated pathogenicity loci.

All 12 convergent isolates contained antimicrobial resistance plasmids (Fig. 3 and Supplementary Data 3). Six had ESBL genes on plasmids, while 4 others had them in their chromosomes (Fig. 3 and Supplementary Data 3). Carbapenemase genes were found on plasmids from six convergent isolates (Fig. 3 and Supplementary Data 3). Altogether, 10 convergent isolates had plasmids that contained genes encoding ESBLs or carbapenemases. OC1092 carried plasmids that contained both a carbapenemase ($bla_{KPC-3}$) and an ESBL ($bla_{SHV-12}$) gene. Only ARLG-4803 lacked plasmids containing these genes (Supplementary Data 3). ARLG-4803, was carbapenem-resistant despite lacking a carbapenemase gene (Supplementary Data 3). However, it did contain a $bla_{CTX-M-15}$ ESBL gene in its chromosome (Supplementary Data 3).

## In vitro phenotypes of convergent isolates

Hypervirulent isolates frequently are resistant to complement-mediated killing in pooled human serum and have mucoid and highly viscous capsules (hmv)[15]. We measured these phenotypes in our 12 convergent isolates and compared them to those of typical hvKP ($n = 11$), MDR-cKP ($n = 11$), and relatively antibiotic-susceptible cKP (NON-MDR-cKP, $n = 6$) isolates (Fig. 5 and Supplementary Data 2). The convergent isolates, along with the MDR-cKP isolates, exhibited a large range in serum sensitivity, perhaps due to diversity in capsule and o-antigen types (Fig. 5A and Supplementary Data 2). However, the differences in median serum resistance between convergent isolates and the other groups were not statistically significant (Fig. 5A). Convergent isolates produced slightly more capsule than MDR-cKP and NON-MDR-cKP isolates and slightly less than hvKP isolates, although these differences were not statistically significant (Fig. 5B). As expected, hvKP isolates made more capsule than MDR-cKP and NON-MDR-

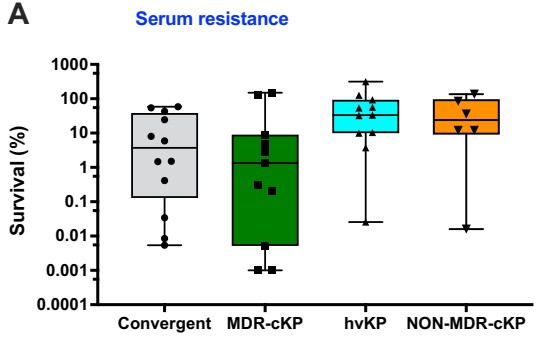

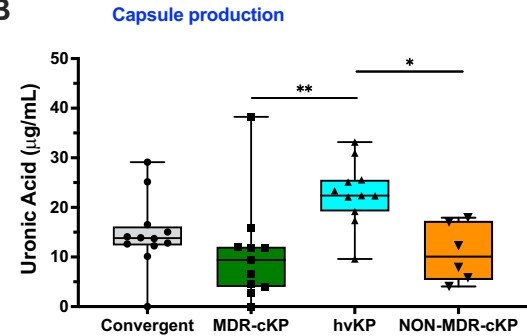

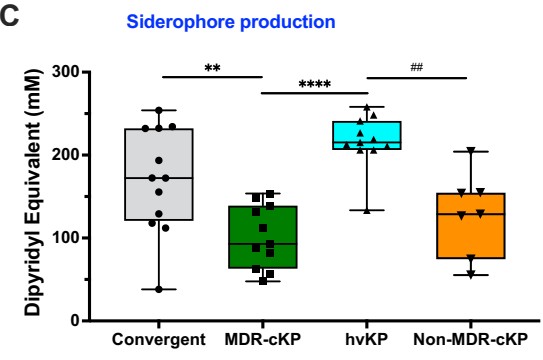

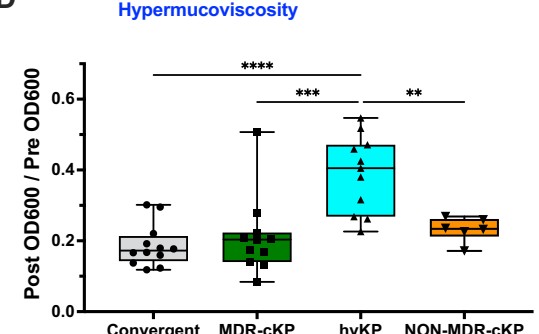

**Fig. 5 | Phenotypic comparison of convergent, MDR-cKP, hvKP, and NON-MDR-cKP isolates.** **A** Survival in pooled human serum, **B** capsule production, **C** siderophore production, and **D** hypermucoviscosity as measured by resistance to sedimentation. Each symbol represents an isolate ($n = 40$). The middle lines represent medians, the boxes represent the interquartile range, and the outer bars represent the minimum and maximum. Data for each isolate are the means of at least 2 independent experiments consisting of 3 technical replicates each. Statistical analysis was performed using one-way ANOVA; **B** *$p = 0.0235$, **$p = 0.0045$, **C** **$p = 0.0076$, ##$p = 0.0029$, ****$p \leq 0.0001$, **D** ****$p \leq 0.0001$. Source data are provided as a Source data file.

cKP. To measure the hypermucoviscosity of these capsules, we performed centrifugation assays. As a group, convergent isolates were no more hypermucoviscous than MDR-cKP and NON-MDR-cKP isolates, whereas hvKP isolates were significantly more hypermucoviscous than the other isolates (Fig. 5D). These results were mostly consistent with a string test analysis to measure hmv; only one convergent isolate (K014) had a positive string test, whereas all hvKP isolates did (Fig. S2B and Supplementary Data 2). Interestingly, some string-test-negative K1 isolates were as resistant to sedimentation by centrifugation as string-test-positive isolates with other capsule types (e.g., TK77 and DHQP1701672) (Fig. S2A). These findings show that convergent isolates more closely resemble cKP isolates than hvKP isolates in capsule-related phenotypes.

Two convergent isolates, KPN226 and DHQP1701672, were hmv-negative despite containing *rmpADC* loci that usually confer an hmv colony morphology. Upon closer inspection, we found that KPN226 contained an early stop codon mutation in *rmpA* that likely disrupted its function. DHQP1701672 contained SNVs immediately upstream of *rmpA* and *rmpD* that may have altered their expression. Alternatively, DHQP1701672 resisted sedimentation to a degree consistent with other hmv+ isolates (Fig. S2A), suggesting that the string test did not adequately measure the hypermucoviscosity of this isolate. One MDR-cKP and three NON-MDR-cKP isolates were also hmv (Fig. S2B), consistent with other reports that cKP isolates may also be hmv[23].

It has been suggested that high levels of siderophore production may account for hypervirulence[43,44]. We therefore measured total siderophore production by the four sets of isolates. Convergent isolates secreted significantly more siderophores than MDR-cKP or NON-MDR-cKP but similar amounts as hvKP isolates (Fig. 5C). Among the 12 convergent isolates, 11 contained *iuc* loci: 6 contained *iuc*1 (an hvKP-associated lineage) and 5 contained *iuc*5 (an *E. coli*-associated lineage) (Supplementary Data 2). Convergent isolates secreted variable amounts of siderophores, and several isolates of both types (*iuc*1 and *iuc*5) secreted amounts of siderophores comparable to hvKP isolates (Fig. S3). Overall, 20 (87%) of the 23 *iuc*+ isolates secreted more siderophores than the *iuc*- isolates, suggesting that the *iuc* locus makes a substantial contribution to total siderophore production. *iuc*1 and *iuc*2 loci were present in hvKP isolates and were consistently associated with high levels of siderophore production (Fig. S3)[11]. Overall, having an *iuc* locus was associated with an increase in siderophore production (two-way ANOVA; $p < 0.001$), but no statistically significant differences were observed in siderophore production between isolates with different *iuc* lineages (Tukey's HSD; $p > 0.05$). These findings indicate that convergent isolates are capable of secreting large amounts of siderophores, and that this phenotype is not linked to a specific aerobactin lineage.

## Virulence of convergent isolates

Although pathogenic genes and in vitro phenotypes are useful markers, the true virulence of a bacterial strain is arguably reflected by its capacity to cause disease in a relevant animal model. For this reason, we quantified the virulence of the 12 convergent isolates using a mouse model of pneumonia by measuring 50% lethal dose ($LD_{50}$). Convergent isolates had a median $LD_{50}$ value of ~$10^8$ CFU, nearly identical to that of MDR-cKP and only slightly more (i.e., less virulent) than NON-MDR-cKP (Fig. 6). In contrast, hvKP isolates had a median $LD_{50}$ value of ~$10^3$ CFU, significantly lower (i.e., more virulent) than that of convergent isolates. Only one convergent isolate, K014, had an $LD_{50}$ value comparable to hvKP isolates ($10^{4.2}$ CFU) (Fig. S4A). K014 is an ST268 isolate collected from a patient with meningitis at NMH in 2016. This isolate, which was

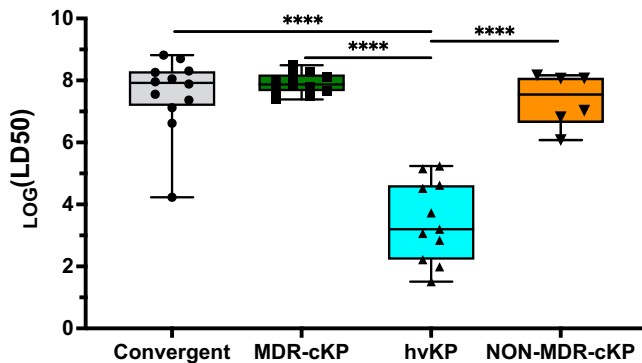

**Fig. 6 | Virulence comparison of convergent, MDR-cKP, hvKP, and NON-MDR-cKP isolates.** Virulence in a murine model of pneumonia was measured for each of the 40 representative isolates. Each symbol represents an isolate ($n = 40$). The middle lines represent medians, the boxes represent the interquartile range, and the outer bars represent the minimum and maximum. Number of mice and dosing used to determine $LD_{50}$ values are included in Supplementary Data 5. Statistical analysis was performed using one-way ANOVA; ****$p \leq 0.0001$. Source data are provided as a Source data file.

resistant to third-generation cephalosporins, contained *rmpADC, iuc*, and *iro* loci, and is the only convergent isolate that was hmv. Interestingly, the CDC ST23 carbapenem-resistant isolate (DHQP1701672) was not hypermucoviscous (by string test) and had an $LD_{50}$ value of $10^{6.6}$, over four logs higher (i.e., lower virulence) than that of other ST23 isolates, despite containing a complete pK2044-like plasmid with intact genes in the *rmpADC, iro*, and *iuc* loci (Figs. 4A, S4B). Thus, most convergent isolates had relatively low virulence levels despite containing virulence genes that are associated with hypervirulence.

## Discussion

Historically, *K. pneumoniae* has existed as two distinct groups: cKP and hvKP. However, recent isolates have emerged with features of both cKP and hvKP, raising the specter of highly invasive strains that are resistant to treatment. Our findings are reassuring and suggest that convergent isolates remain rare in the U.S. Furthermore, although our sample size was small, most of the examined convergent isolates did not have enhanced virulence, as measured by a mouse model of pneumonia. Of note, though, one ESBL-producing isolate did have virulence comparable to typical hvKP isolates, indicating that vigilance for convergent isolates must remain a priority (Fig. 6).

Despite a variety of combinations of virulence genes and strain backgrounds in our 12 convergent isolates, only one had an $LD_{50}$ value comparable to hvKP strains (Fig. 5D). These observations are consistent with those of Martin and colleagues, who described convergent ST147 and ST307 strains from Italy and Germany, respectively, containing *iuc, rmpADC*, and *rmpA2* loci[32]. Both strains had unexpectedly attenuated virulence in a mouse subcutaneous infection model, similar to a cKP control strain. However, our collection is small and did not contain representatives of strains recently linked to severe human infections in Asia[33]. It remains possible that interrogation of a larger and international collection of convergent isolates would yield a higher proportion of virulent convergent isolates.

There are several potential explanations for the low virulence of our convergent isolates: (1) It is possible that the aerobactin biosynthesis locus may not be sufficient to enhance virulence to the level of hvKP strains. (2) It is conceivable that the *E. coli* aerobactin lineage (*iuc5*) confers a lower level of virulence than *iuc1* or *iuc2*, which are usually present in hvKP strains. (3) Convergent isolates exhibit diverse capsule types, some of which may offer lower resistance to phagocytosis or complement compared to hvKP-associated KL1 and KL2 types[45]. (4) The proteins encoded by the *rmpADC* operons

located on plasmids may fail to interact with their cognate chromosomal targets in strains with capsule types or sequence types not typically associated with hvKP. (5) Convergent isolates may have acquired mutations in important pathogenicity genes, as has been reported by others[34]. In our convergent isolates, we observed mutations in the *rmpA* and *rmpA2* genes. In addition, OC1092 had a frameshift mutation in a gene (*wzi*) necessary for capsule production. Additional studies and examination of larger numbers of convergent isolates may uncover more reasons for the low virulence of many convergent strains.

Only one (K014) of our 12 convergent isolates was hmv, and this isolate was also the only isolate with high virulence in the mouse model. These findings suggest hmv contributes to the overall virulence of convergent isolates and are consistent with one possible explanation for the attenuation of convergent isolates. Since the evolution of convergent isolates presumably requires acquisition of mobile elements (containing either antibiotic resistance or virulence genes), it is possible that this evolution is restricted by hmv itself. The thick capsules associated with hmv may prevent the transfer of mobile elements, limiting the evolution of convergent isolates to either donor or recipient hvKP strains that have lost the ability to be hmv. In the former case, the defective hvKP donor strain transfers a virulence plasmid containing mutated activators of hmv to an antibiotic-resistant strain, and in the latter case, the defective recipient hvKP strain receives an antimicrobial resistance (AMR) plasmid from an antibiotic-resistant strain. In both cases, the resulting convergent strain is not hmv and presumably not highly virulent. In agreement with this hypothesis, most convergent isolates in our study acquired only siderophore-encoding fragments of virulence plasmids that did not confer the hmv phenotype. Most of those few convergent isolates that acquired AMR plasmids (e.g., DHQP1701672) presumably had mutated to become non-hmv recipients. However, it is important to note that one of our isolates (K014) was indeed hmv, highly virulent, and highly antibiotic resistant, highlighting the fact that if such plasmid transfer barriers do exist, they can be overcome.

Although not an epidemiological study, our results suggest that convergent isolates remain rare in the U.S. Of the 2608 highly antibiotic-resistant *K. pneumoniae* isolates we screened, only 1.8% contained hvKP-associated genes. This proportion is less than the proportions observed in other studies within the U.S., likely because we focused on collections of antibiotic-resistant isolates rather than bloodstream or liver abscess isolates, which would be expected to be enriched for virulence genes[7,10,46,47]. Our prevalence of 1.8% was also lower than the 7.3% prevalence of convergent isolates observed in South and Southeastern Asia bloodstream isolates, perhaps because the higher prevalence of hvKP isolates in Asia facilitates the emergence of convergent strains[29].

hvKP strains produce large amounts of siderophores, which may contribute to their ability to thrive in the iron-limited host environment[48]. Here, we confirm that hvKP isolates produce significantly more siderophores than MDR-cKP or NON-MDR-cKP isolates. Interestingly, convergent isolates also produce significantly more siderophores than MDR-cKP or NON-MDR-cKP isolates, and four convergent isolates (ARLG-3346, K014, OC1092, and KPN1402) produced more siderophores than the median amount produced by hvKP isolates. Among the different siderophores, aerobactin is thought to play a major role in hvKP virulence and is responsible for the high amounts of siderophores produced by some hvKP strains[48,49]. Although 11 of the 12 convergent isolates we tested had *iuc* loci (and 6 had the hvKP-associated *iuc1* loci), only one (K014) was highly virulent in mice. This suggests that either these *iuc* loci are non-functional and do not produce substantial amounts of aerobactin or that aerobactin production by itself is not sufficient to confer high levels of virulence in mice. In either case, our findings suggest that the presence of aerobactin biosynthesis genes alone should not be used in future studies to make

inferences about the virulence of *K. pneumoniae* strains. Rather, the total number of virulence factors should be considered when attempting to predict strain virulence. We also emphasize that our study examined *associations* between pathogenic determinants and virulence; additional genetic studies will be necessary to determine which determinants play a *causal* role in virulence.

Our study has several limitations. First, we examined isolates from only parts of the U.S., thus our results may not be representative of those occurring in other parts of the world. For example, ST11 and ST23 convergent isolates were rare in our study but are common in parts of Asia[29]. Second, we screened large collections of antibiotic-resistant cKP for virulence genes; the approach of screening isolate collections from clinically severe or invasive infections for hvKP-associated genes and antibiotic-resistance genes may yield different types of convergent isolates[25]. Third, we examined only a relatively small number of convergent isolates of limited types for virulence in mice. Nonetheless, our results may serve as a foundation for larger studies in the future. Fourth, we used a mouse model of pneumonia to quantify virulence, and it is possible that other infection models (e.g., intraperitoneal inoculation) may yield different results. However, the pneumonia model has been widely used to quantify the virulence of hvKP strains and nicely distinguishes hvKP from cKP[11,50]. Fifth, we cannot exclude the possibility that convergent isolates acquired virulence-attenuating mutations during passaging in our laboratory. However, we minimized the numbers of passages each isolate underwent, and we did not observe attenuation in the 11 similarly passaged hvKP isolates used as comparators. For these reasons, we feel this possibility is less likely. Finally, firm conclusions on the clinical risk associated with convergent isolates requires correlations with clinical outcomes in patients from which these isolates were cultured.

In conclusion, most of the convergent isolates in our study were not as highly virulent as hvKP strains. While containing these additional virulence factors is likely to pose an increased clinical risk in humans, our data suggests that these isolates are unlikely to cause the type of infections typically described for hypervirulent strains[36]. These findings suggest that additional studies should be performed to clarify whether convergent strains are indeed more virulent than cKP in mouse and human infections. However, strains that are both highly virulent and highly antibiotic resistant do occur, highlighting the need for continued study of convergent isolates.

## Methods

### Ethical regulations
Experiments using *K. pneumoniae* isolates were conducted in compliance with the Northwestern Institutional Biosafety Committee. All animal procedures were conducted in accordance with the Northwestern University Animal Care and Use Committee (protocol IS00002172).

### Bacterial isolates and growth conditions
This study used three distinct collections of drug-resistant *K. pneumoniae* isolates collected from U.S. hospitals (Supplementary Data 1). In addition, typical MDR-cKP, NON-MDR-cKP, and hvKP isolates used for comparisons with convergent isolates are listed in Supplementary Data 2. The group of typical hvKP isolates included canonical hypervirulent STs (ST23, ST86, ST66, and ST380) that each contained a large virulence plasmid and was hmv (Supplementary Data 2). In addition, strain KPPR1 was added to the hvKP group because it is commonly used to study hvKP[51]. The MDR-cKP group included typical antibiotic-resistant high-risk clones (e.g., ST258, ST11, ST16, ST15, and ST14) that did *not* contain *iucA*, *iroB*, *rmpA*, or *rmpA2* genes. The NON-MDR-cKP group included 5 relatively antibiotic susceptible cKP clinical isolates. In this last group, we included for comparison an hvKP isolate, hvKP5, that had been cured of its virulence plasmid (designated TK77; Supplementary Data 2).

Selected bacterial isolates were recovered from frozen stocks. Bacteria were grown at 37 °C in lysogeny broth (LB) or on LB agar plates.

### Genome sequences
Illumina sequencing reads were obtained from NCBI for bioprojects PRJNA658369 (CRACKLE2), PRJNA514245 (Houston-Methodist) and PRJNA788509 and PRJNA395086 (NMH)[10,37–39,52]. Reads were downloaded from NCBI using Sratoolkit v2.11.1. For new strains sequenced as part of this study, a single colony was cultured overnight at 37 °C in LB and used to prepare genomic DNA using the Maxwell 16 system (Promega Corp., Madison, WI, USA). To generate paired-end reads of either 300 bp or 150 bp, Illumina sequencing libraries were created using either the Nextera XT kit (Illumina, Inc., San Diego, CA) or the Seqwell kit (Seqwell, MA, USA). The libraries were then sequenced on an Illumina MiSeq or NextSeq 500 instrument. Reads were trimmed using Trimmomatic (v0.36)[8,53]. De novo assembly was performed using SPAdes (v3.9.1)[53,54]. Contigs were filtered based on coverage (minimum 5x) in trimmed reads using a custom perl script.

Nanopore sequencing was performed according to previously described protocols[10,55–58]. Libraries were prepared using the ligation sequencing kit (SQK-LSK109 or SQK-LSK114, Oxford Nanopore, UK) and sequenced on either a MinION or GridION instrument using either a FLO-MIN106 or FLO-MIN114 flow cell. Sequence reads were demultiplexed and base-called using Guppy (v3.4.5)[59]. Base-called reads were filtered using Filtlong (v0.2.1) removing any reads shorter than 1000 bp. Assemblies were completed using the Trycycler pipeline (v0.5.4)[60]. Filtered reads were subsampled and assembled using Flye (v2.9), Unicycler (v0.5.0), and Raven (v2.3)[61–63] Generated contigs were clustered, reconciled, and consensus sequences were generated. Consensus sequences were polished using long-reads (Medaka v1.9.1) and short-reads (Polypolish v0.5.0 and Polca as part of the MaSurCA package v4.1.0) (Supplementary Data 4)[64,65]. Finally, annotation was performed using the NCBI Prokaryotic Genome Annotation Pipeline[66].

### Sequence analysis
Assembled whole-genome sequences were evaluated for multilocus sequence type, capsule locus type, antimicrobial resistance gene content and the presence of virulence genes using the bioinformatics tools Kleborate v2.1 and Kaptive v0.7.3 with default settings[42,67]. BLAST Ring Image generator (BRIG) was used to align plasmids with a sequence similarity threshold of 85%[68]. MASH (v2.3) was utilized to cluster plasmids based on Jaccard index[69]. Cytoscape (v3.8.2) was used to create plasmid networks. MOB-suite (v3.1.7) was used to identify plasmid replicons[70]. Maximum-likelihood phylogenetic trees were constructed using FastTree 2 (v4.0.3) based on the core genome[71]. The core genome was defined as sequences present in 95% of isolates[71]. Identification of single nucleotide polymorphisms (SNPs) was performed by aligning raw Illumina reads to the genome of the reference strain NTUH-K2044 using bwa-0.7.15. Mashtree (v1.4.3) was used for clustering of individual plasmids[72,73]. The phylogenetic trees were visualized and annotated using iTOL (v4)[74].

### Hypermucoviscosity testing
Hypermucoviscosity was measured by string test and by centrifugation. The string test was performed as described previously[75]. Briefly, isolates were grown overnight at 37 °C on LB agar. A colony was lifted with a loop to evaluate the formation of a viscous string between the loop and the colony. A positive string test was defined as a string length ≥5 mm. For centrifugation assays, isolates were cultured overnight on LB agar at 37 °C from frozen stocks. Multiple colonies were picked and cultured overnight in LB at 37 °C with shaking. Cultures were centrifuged at $2000 \times g$ for 5 min, and the optical density at 600 nm ($OD_{600}$) of the supernatant was measured. Data are presented as the $OD_{600}$ post-centrifugation divided by $OD_{600}$ pre-centrifugation.

## Serum survival

Pooled, flash-frozen human serum (Innovative Research) was stored at −80 °C. Serum was thawed at room temperature. A portion of the human serum was heat-inactivated by incubation at 55 °C for 1 h. *K. pneumoniae* were grown overnight in 5 mL of LB and then subcultured in fresh LB to an $OD_{600} = 2.5 \pm 0.1$. Bacteria were then pelleted by centrifugation at $3041 \times g$ for 20 min. The bacterial pellets were resuspended in 1 mL PBS and adjusted to $2 \times 10^6$ CFU/mL. Then 100 μL of bacterial suspension was added to 900 μL of active or heat-inactivated human serum and gently vortexed. Experiments were performed in triplicate and bacteria were serial diluted and enumerated following 0 and 180 min of incubation at 37 °C.

## Siderophore detection

Siderophores were quantified by the Chrome-azurol S assay[76]. Briefly, 5 mL aliquots of LB medium were inoculated with single colonies of bacteria and grown overnight with aeration at 37 °C. One mL aliquots of overnight cultures were harvested by centrifugation (2 min, $21,130 \times g$), washed twice with M9 minimal medium supplemented with glucose, and resuspended in 1 mL M9 minimal medium[76]. A total of 50 μL of bacterial resuspension was inoculated into 5 mL of fresh M9 minimal medium and grown for approximately 7 h at 37 °C with aeration to an $OD_{600} \sim 1.3$[76]. After centrifugation at $4300 \times g$ for 20 min at 4 °C, supernatant was carefully collected and passed through a 0.22 μm filter. Next, 150 μL of supernatant was mixed with 150 μL of CAS shuttle solution[76]. In addition, serially diluted concentrations of the iron chelator dipyridyl in M9 medium were used to generate a standard curve. The plate was incubated at room temperature for 15 min in the dark, and absorbance was measured at $OD_{630}$ nm using a BioTek plate reader.

## Capsule quantification

*K. pneumoniae* capsule was purified[51]. Briefly, bacteria were cultured overnight in LB medium and then subcultured to an $OD_{600} = 2.5 \pm 0.1$ in fresh LB. A total of 500 μL was combined with 100 μL of 1% Zwittergent 3-14 in 100 mM citric acid buffer, pH 2. This mixture was incubated at 50 °C for 30 min with occasional mixing. Bacterial cells were pelleted by centrifugation at $17,000 \times g$ for 2 min, after which 250 μL of supernatant was combined with 1 mL of absolute ethanol (final concentration of 80%) and placed on ice for 30 min. Suspensions were cleared by centrifugation at $17,000 \times g$ for 5 min, and the supernatant was decanted. Pelleted samples were incubated at 37 °C for 30 min or until the precipitates were dry. Then, 200 μL of ddH2O was added, and the precipitate was allowed to redissolve for 1 h at 37 °C. A total of 1 mL of 12.5 mM sodium tetraborate in concentrated sulfuric acid was added, and the samples were boiled for 5 min at 100 °C and allowed to cool to room temperature. Next, 20 μL of 0.15% 3-hydroxydiphenyl in 0.5% sodium hydroxide was added; 20 μL of 0.5% sodium hydroxide was added to the control set for a background measurement. Absorbance was read at 520 nm.

## Mouse studies

Six- to eight-week-old C57BL/6 female mice were purchased from Jackson Labs and infected intranasally[10]. Briefly, *K. pneumoniae* was recovered from frozen stocks and grown overnight on LB agar at 37 °C and then subcultured in LB medium at 37 °C with shaking for 3 hours the day of infection. Bacteria were washed 3 times in PBS, resuspended in PBS, and then diluted with PBS to the indicated doses by measuring $OD_{600}$. Inocula sizes were confirmed by plating bacteria for colony enumeration. Mice were anesthetized by intraperitoneal injection with a mixture of ketamine (100 mg/kg) and xylazine (20 mg/kg). A total of 50 μl of *K. pneumoniae* suspension was placed on the nares of mice (25 μl per nare) to allow for aspiration into the lungs. Mice were monitored for pre-lethal illness over the subsequent two-week period. Mice were humanely euthanized and scored as dead when they met pre-determined criteria: >20% weight loss, abnormal breathing, or a hunched posture with limited movement. The experimenters were blinded to the strain and dose used. The $LD_{50}$ was calculated from doses and mouse outcomes (development of pre-lethal illness) using the R package drc[77]. Detailed data, including the strain dose, number of deaths, and total mice infected, are provided in Supplementary Data 5. Sex was not included in the study design as previous experiments found no difference in *K. pneumoniae* infections among male and female mice. All procedures were conducted in accordance with the Northwestern University Animal Care and Use Committee (protocol IS00002172). Mice were housed at the Northwestern Center for Comparative Medicine at Temperatures of 65–75 °F (-18–23 °C) with 40-60% humidity with a 12:12 light dark cycle.

## Statistics and reproducibility

Statistical analyses were done using One-Way ANOVA as part of GraphPad Prism 10. No statistical method was used to predetermine sample size. No data were excluded from the analyses. Groups of mice were determined at random and investigators were blinded to the strains paired with each group of mice.

## Reporting summary

Further information on research design is available in the Nature Portfolio Reporting Summary linked to this article.

## Data availability

The whole-genome assemblies of bacteria sequenced for this study have been deposited at GenBank under BioProject number PRJNA788509. Assembly accession numbers are included in Supplementary Data 4. Kleborate data are included in Supplementary Data 1. Phenotypic data for individual strains are found in Supplementary Data 2. There are no restrictions on data availability. Source data are provided with this paper.

## Code availability

Code used in this work is available on github.com/tkochan.

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

## Acknowledgements

This work was funded by American Heart Association grant 837089 (T.K.), Chicago Biomedical Consortium Catalyst Award (A.H., Z.B.), and National Institute of Health grants T32 AI007476 (T.K.), R01 AI118257 (A.H.), R21 AI153953 (A.H.), K24 AI104831 (A.H.), R21 AI164254 (A.H.), U19 AI35964 (A.H.), T32 AI095207 (R.M., A.H.), R01 AI173064 (Z.P.B., A.H., E.O.) The funding sources had no influence on the design of the study and collection, analysis, interpretation of data, or writing of the manuscript.

## Author contributions

T.K. and A.H. designed the study. T.K., S.N., B.C., A.V., M.V., M.L.C., S.M., C.A., T.W., J.N., E.V., and J.O.M. performed the phenotypic experiments. T.K. and E.O. performed the computational analyses. R.M. and B.C. analyzed and interpreted the patient data. T.K., S.N., S.M., E.O., A.V., and A.H. initially analyzed the data. D.D., L.C., B.K., S.W.L., J.M.M., Z.B., and R.W. contributed to interpretation of the data. T.K. and A.H. wrote the paper, and all other authors contributed to the writing. All authors read and approved the manuscript.

## Competing interests

The authors declare no competing interests.

## Additional information

[1]Laboratory of Respiratory and Special Pathogens, Division of Bacterial, Parasitic, and Allergenic Products, Office of Vaccines Research and Review, Center for
Biologics Evaluation and Research, Food and Drug Administration, Silver Spring, MD, USA. [2]Department of Microbiology-Immunology, Feinberg School of
Medicine, Northwestern University, Chicago, IL, USA. [3]Division of Infectious Diseases, Department of Medicine, Feinberg School of Medicine, Northwestern
University, Chicago, IL, USA. [4]Division of Infectious Diseases, University of North Carolina, Chapel Hill, NC, USA. [5]Center for Discovery and Innovation,
Hackensack Meridian Health, Nutley, NJ, USA. [6]Department of Pathology and Genomic Medicine, Houston Methodist Hospital Research Institute, Houston,
TX, USA. [7]College of Pharmacy, University of Illinois at Chicago, Chicago, IL, USA. [8]Division of Pulmonary and Critical Care Medicine, Department of Medicine,
Feinberg School of Medicine, Northwestern University, Chicago, IL, USA. [9]Simpson Querrey Institute for Epigenetics, Feinberg School of Medicine, North-
western University, Chicago, IL, USA. ✉e-mail: Travis.Kochan@fda.hhs.gov

