## [Peer Review File · Nature Communications]

REVIEWER COMMENTS

Reviewer #1 (Remarks to the Author):

This is an overall high-quality manuscript describing and interesting study of 'convergent' *Klebsiella pneumoniae* harbouring both drug-resistance and virulence genes.

My main concern is that the authors seek to overturn the idea that strains harbouring these virulence genes pose an increased clinical risk to humans, on the basis of a small number of experiments (n=12 strains) and a single animal model (assessed with a single readout, being lethal dose).

1. No details are provided of the microbiological culture steps, so how do we know the isolate used to inoculate mice carries the same genes as the isolate from which DNA was extracted? As I understand it the authors are using sequence data obtained some time ago; they used this sequence data as the basis to identify convergent isolates to study experimentally; then they go back to stored isolates, revive and passage them, and start experiments. It has been noted that hmv and siderophore synthesis genes can be negatively selected during growth in culture in rich media, as they impose a fitness cost under these conditions. So one concern here is that virulence genes could be mutated, or virulence plasmids lost, during culture to grow up the strains for inoculating mice. No details are given about how isolates were stored, or what the culture history looks like. Was the same passage used to culture strains for experimental assays and replicate inoculations? Does each set of inoculations represent a unique biological replicate in terms of independent passage of the isolate from original stock? Was there any resequencing done at any stage to ensure the genome has not mutated?

2. This reviewer's interpretation of the current literature on virulence genes is that nobody really proposes that a single locus (e.g. *iuc* OR *rmp* OR *clb*) alone turns a KP strain into a hypervirulent strain (i.e. changing the lethal dose from 10^8 to 10^3 in this pneumonia model). I would instead expect that a combination of bacterial factors, including specific K types, would be required for this, which is indeed what the authors found here (the one strain that had low lethal dose had the full suite of virulence factors, as per Table 2). However (1) this does not mean that the individual genes/loci do not modify severity in any clinically meaningful way (i.e. these genes may still modify clinical risk); and (2) it does not negate the idea that the public health risk associated with such strains is greater than strains that lack any hv loci... if the genetics are indeed combinatorial, then a strain with some acquired hv virulence factors is a shorter evolutionary step away from acquiring 'the full set' of hv virulence factors and therefore has a greater risk potential. I would like to see some deeper discussion about what these findings actually mean for interpretation of genomic data, as to my mind it doesn't really change the current interpretation.

Minor comments

Line 61 - “capsule types (KLs)” should be “capsule locus types (KLs)” as KL refers to the locus not the produced antigen/serotype per se

Line 63 - please define ‘multidrug resistant (MDR)’ as there are different ways to define this for Kp. E.g. Magiorakos et al 2014 define MDR for Enterobacteriaceae as resistance to drugs in at least three antimicrobial classes; however because all Kp have expected resistance to ampicillin, MDR in Kp is sometimes defined as resistance to at least three drug classes in addition to ampicillin. It is stated in the next line that MDR cKP produce ESBLs, do the authors mean that is part of their definition of MDR?

Line 98 - This paragraph introduces the emergence of convergent strains with both MDR and hv related genes, and the lack of evidence that these are actually able to cause clinically more severe infections. Surprisingly, no mention is made of the 2017 Lancet Microbe report of an outbreak of KPC-producing ST11-KL47 strains in China which killed all 5 patients infected (doi: 10.1016/S1473-3099(17)30489-9)

There are other studies from China (where hv/CRKP are very common) that have explored mouse lethality and/or clinical severity of such strains, this is highly relevant to the topic of the present manuscript and should be included in the Introduction and Discussion.

Some examples:

- i) “Hypervirulent carbapenem-resistant *Klebsiella pneumoniae* causing highly fatal meningitis in southeastern China” (doi: 10.3389/fpubh.2022.991306), report significantly greater mortality in patients with hv-CRKP vs nonhv-CRKP meningitis
- ii) “Nosocomial dissemination of hypervirulent *Klebsiella pneumoniae* with high-risk clones among children in Shanghai” (doi: 10.3389/fcimb.2022.984180), divide isolates carrying hv genes into those that show hv phenotype in vitro/in vivo and compare clinical characteristics of the patients
- iii) “Molecular epidemiology of carbapenem-resistant hypervirulent *Klebsiella pneumoniae* in China” (doi: 10.1080/22221751.2022.2049458), tested 784 CRKP for virulence genes and hypermucoidy, found only 13% of rmp+ strains were actually hypermucoid (mostly due to mutation of the rmpA gene), these were also virulent in a neutrophil assay although unfortunately they do not report clinical severity in patients
- iv) “Genomic epidemiology study of *Klebsiella pneumoniae* causing bloodstream infections in China” (doi: 10.1002/ctm2.624), report clinical results for n=239 blood stream infections with Kp including 129 with hv genes of which 110 were hv in a Galleria model; hv strains were associated with increased CRP, neutrophil counts, total white blood cells; other clinical variables showed potential associations also

Line 120 - why define the term “highly antibiotic resistant” that is based on resistance to a single drug class? Why not just refer to these directly as third-generation cephalosporin resistant (3GCR), and avoid the need to define a term that obscures what is meant? This would be much clearer for readers

Line 146 - the abbreviation “CPE” is typically used to delineate carbapenemase-producing Enterobacteriaceae, I find it confusing to redefine this as ‘carbapenemase’; why not just use the word ‘carbapenemase’?

Lines 144-149 - Again it is not clear to me what is meant by ‘MDR Kpn’. In the intro it was stated that MDR Kpn carry ESBLs; but here it is stated that 90% carry resistance determinants for ≥ 3 drug classes; 78.6% carried an ESBL, others had a carbapenemase but some lacked both ESBL and carbapenemase... so what is the definition of MDR here? Clearly it is neither ‘ ≥ 3 drug classes’ nor presence of ESBL or carbapenemase, so how is it defined?

Line 150 - typo, ‘atleast’ should be ‘at least’

Lines 308-309 state that OC1092 had a frameshift in wzi and was presumably less virulent for that reason, please clarify was it confirmed whether this strain produces capsule?

Methods: please add version numbers for all software

Methods: The ‘Mouse studies’ section lacks detail on how lethal dose was determined (and also lacks critical details on bacterial passage, as noted above)

Paragraph beginning line 235 - only one of the convergent isolates was hmv; this is one of three isolates with a rmpADC locus noted in Table 2. Was the locus intact in the other 2 strains (ST23 DHQP1701672 and ST16 KPN226)? (The other 9 isolates would not be expected to be hmv as they lack the rmp locus.) Was this checked in the culture from which DNA was extracted?

Reviewer #2 (Remarks to the Author):

This important study by Kochan et. al concerns with the convergence of the two pathotypes of *Klebsiella pneumoniae* (Kp); namely classical Kp strains that are multidrug-resistant (MDR) and hypervirulent

strains that generally have a virulence plasmid but tend to be antibiotic sensitive. Recent studies have suggested that there is convergence of Kp pathotypes with the identification of strains that are MDR but also have hvKp characteristics. Here they look at MDR Kp strains in the US from different sites (2608 strains in total), and through sequencing identify 47 strains (plus the CDC isolate) that have genetic characteristics of hvKP. They identified that most of them have the iuc locus, a hallmark of hvKp strains that either belonged to E. coli lineage or the hvKp lineage. From their data it appears that MDR strains had acquired the hvKp genetic determinants. Surprisingly, from the narrowed list only one strain appeared to have the hmv phenotype. Furthermore, no differences in capsule amount, serum sensitivity were detected, and the convergent strains had similar virulence to cKP MDR strain, even though they produced more siderophores. Overall the study is logical and well written, even though it is primarily a descriptive work, its importance can be derived from the clinical relevance of convergent strains.

1. It is surprising to the reviewer that even with having the rmp locus the convergent strains are hmv negative. Is the rmp locus being expressed? Comparison could be to an hvKp isolate. Furthermore, did the authors determine if there were any mutations in the capsule locus, specifically in wzc, to which RmpD was recently shown to interact (Ovchinnikova 2023)? This would also provide insight in to the hmv negative phenotype. Also, in their uronic acid assay, are the samples adjusted to OD600? As absorbance can be affected by capsule levels, adjustment to CFU count should be considered.

2. The string test performed is highly qualitative. As it is only 12 strains that were further selected for testing, the authors should perform the spin down hmv assay as described by Walker et al 2020, which is a more stringent and quantitative assay for hmv phenotype.

3. It is not clear to the reviewer if they observed any differences in their serum killing between different isolates? Which is surprising as Pomakova et al 2011 and other studies have shown that hvKP isolates have higher resistance. For all the panels in Figure 5, the reviewer would suggest that they show individual strains as a bar graph, as it is important to see if there is a correlation between any of the tested strains phenotypes and their genotypes. (An example would be Russo et al 2021) By pooling everything it dilutes the results. They can still show the pooled data as well.

4. A concern to the reviewer is in regards to strain(s) site of isolation. It appears that ~50% of the isolates were from urine compared to other sites of infection. This could bias their findings. This should be addressed in the discussion section. It would also be useful to list the site of isolation for the tested strains, rather than have the reader dig through supplemental data to figure out the site of isolation for the 12 tested strains.

Minor point.

a. In line 148 should that also include carbapenems?

We thank the reviewers for their insightful comments, which have helped us improve the manuscript. Below, we address each concern in turn.

Reviewer #1

My main concern is that the authors seek to overturn the idea that strains harbouring these virulence genes pose an increased clinical risk to humans, on the basis of a small number of experiments (n=12 strains) and a single animal model (assessed with a single readout, being lethal dose). **This point is well taken. We have now revised the manuscript to temper our conclusions. We now state that our results suggest that additional studies should be performed to clarify the clinical impact of convergent strains.**

1. No details are provided of the microbiological culture steps, so how do we know the isolate used to inoculate mice carries the same genes as the isolate from which DNA was extracted? As I understand it the authors are using sequence data obtained some time ago; they used this sequence data as the basis to identify convergent isolates to study experimentally; then they go back to stored isolates, revive and passage them, and start experiments. It has been noted that hmv and siderophore synthesis genes can be negatively selected during growth in culture in rich media, as they impose a fitness cost under these conditions. So one concern here is that virulence genes could be mutated, or virulence plasmids lost, during culture to grow up the strains for inoculating mice. No details are given about how isolates were stored, or what the culture history looks like. Was the same passage used to culture strains for experimental assays and replicate inoculations? Does each set of inoculations represent a unique biological replicate in terms of independent passage of the isolate from original stock? Was there any resequencing done at any stage to ensure the genome has not mutated? **These are great points and we appreciate the overall emphasis on rigor and reproducibility.**

With regard to passaging, all biological replicates were taken from a freezer stock stored at -80C. Isolates were streaked on LB agar overnight at 37C and then subcultured in LB broth at 37C. For each biological replicate we went back to the same frozen stock and struck out a new plate. In all experiments and manipulations, passaging was minimized. The Methods have now been updated to include more specific details on microbiological steps.

We did use illumina sequencing data that was acquired previously (the Houston and Crackle2 isolates were sequenced several years ago), but we did repeat sequencing (using the Nanopore platform) on all 40 strains. In particular, repeat sequencing of the 12 convergent isolates utilized for phenotypic analysis showed the same sequence types, capsule types, and virulence gene content as those from the original Illumina sequencing. These findings excluded the possibility of plasmid or gene loss during laboratory manipulations. Ruling out the emergence of SNVs during laboratory growth is more difficult, as Nanopore sequencing is not of sufficient quality to exclude SNVs. However, we did not observe loss of virulence in any of our 11 typical hvKP isolates used as comparators in this study, even though each of these were passaged in the identical manner as the convergent isolates. Likewise, as group these typical hvKP

isolates produced large amounts of capsules and siderophores. Therefore, we think it would be unusual if 11 of the 12 convergent isolates acquired mutations during passaging but none of the 11 typical hvKP isolates acquired mutations during passaging. However, we have now revised the manuscript to mention these considerations and list them as a limitation of our approach.

2. This reviewer's interpretation of the current literature on virulence genes is that nobody really proposes that a single locus (e.g. iuc OR rmp OR clb) alone turns a KP strain into a hypervirulent strain (i.e. changing the lethal dose from 10^8 to 10^3 in this pneumonia model). We agree. However, we feel that there remains a lack of consensus on this issue. As a result, published papers use a variety of different criteria to define hvKP:

<https://doi.org/10.3389/fpubh.2022.991306>: defined as iuc and Peg-344 positive

<https://doi.org/10.3390/microorganisms11030661>: defined as iuc positive

<https://doi.org/10.1128/aac.01127-16>: defined as iuc positive

<https://doi.org/10.1089/mdr.2019.0096>: defined as community-acquired blood stream infections

<https://doi.org/10.3389/fcimb.2022.882210>: defined as positive wzy-K1, ≥ 3 positive siderophore genes (entB, irp2, iroN and iucA), or ≥ 1 positive capsule-regulating genes (p-rmpA2, c-rmpA/A2 and p-rmpA)

<https://doi.org/10.1007/s10096-015-2551-2>: defined as strains positive for p-rmpA and iroB and iucA

Note that several of these papers defined strains as hypervirulent if they contained iuc even in the absence of other typical hvKP virulence loci. Given this lack of consensus, we decided to use a definition of "convergence" that was inclusive of the currently available published literature. However, in the Discussion we state, "In either case, our findings suggest that the presence of aerobactin biosynthesis genes alone should not be used in future studies to make inferences about the virulence of *K. pneumoniae* strains."

I would instead expect that a combination of bacterial factors, including specific K types, would be required for this, which is indeed what the authors found here (the one strain that had low lethal dose had the full suite of virulence factors, as per Table 2). However (1) this does not mean that the individual genes/loci do not modify severity in any clinically meaningful way (i.e. these genes may still modify clinical risk); and (2) it does not negate the idea that the public health risk associated with such strains is greater than strains that lack any hv loci... if the genetics are indeed combinatorial, then a strain with some acquired hv virulence factors is a shorter evolutionary step away from acquiring 'the full set' of hv virulence factors and therefore has a greater risk potential. I would like to see some deeper discussion about what these findings actually mean for

interpretation of genomic data, as to my mind it doesn't really change the current interpretation. We agree with all these points. We have now revised the Discussion section to clarify that our study demonstrates an unexpectedly low level of virulence of a small set of convergent isolates in a mouse model of pneumonia but that additional studies are necessary to assess the clinical risk associated with convergent isolates. In addition, we now emphasize that our study examines only associations, and that genetic studies will be required to demonstrate the roles of individual virulence determinants in the overall virulence of convergent and hvKP isolates.

Minor comments

Line 61 - "capsule types (KLs)" should be "capsule locus types (KLs)" as KL refers to the locus not the produced antigen/serotype per se. The text has been updated as suggested.

Line 63 - please define 'multidrug resistant (MDR)' as there are different ways to define this for Kp. E.g. Magiorakos et al 2014 define MDR for Enterobacteriaceae as resistance to drugs in at least three antimicrobial classes; however because all Kp have expected resistance to ampicillin, MDR in Kp is sometimes defined as resistance to at least three drug classes in addition to ampicillin. It is stated in the next line that MDR cKP produce ESBLs, do the authors mean that is part of their definition of MDR? The text has been updated to define MDR as strains that are resistant to 3 or more classes of clinically relevant antibiotics. Because ampicillin is not used clinically to treat Kp infections (as the reviewer mentioned, all Kp are resistant to it), this revised definition excludes it.

Line 98 - This paragraph introduces the emergence of convergent strains with both MDR and hv related genes, and the lack of evidence that these are actually able to cause clinically more severe infections. Surprisingly, no mention is made of the 2017 Lancet Microbe report of an outbreak of KPC-producing ST11-KL47 strains in China which killed all 5 patients infected (doi: 10.1016/S1473-3099(17)30489-9) We thank the reviewer for pointing out this omission. We now mention this paper in the Introduction and Discussion.

There are other studies from China (where hv/CRKP are very common) that have explored mouse lethality and/or clinical severity of such strains, this is highly relevant to the topic of the present manuscript and should be included in the Introduction and Discussion.

Some examples:

i) "Hypervirulent carbapenem-resistant *Klebsiella pneumoniae* causing highly fatal meningitis in southeastern China" (doi: 10.3389/fpubh.2022.991306), report significantly greater mortality in patients with hv-CRKP vs nonhv-CRKP meningitis. We considered citing this paper in the Introduction, however, note that the "nonhv-CRKP" group in this paper (Fig 2D) contains 3 CR-cKP strains, 6 CS-hvKP strains, and 14 CS-cKP strains making it difficult to interpret the reported mortality comparison.

ii) “Nosocomial dissemination of hypervirulent *Klebsiella pneumoniae* with high-risk clones among children in Shanghai” (doi: 10.3389/fcimb.2022.984180), divide isolates carrying hv genes into those that show hv phenotype in vitro/in vivo and compare clinical characteristics of the patients. **We now cite this study and mention that it suggests higher virulence of convergent isolates in an animal model of infection.**

iii) “Molecular epidemiology of carbapenem-resistant hypervirulent *Klebsiella pneumoniae* in China” (doi: 10.1080/22221751.2022.2049458), tested 784 CRKP for virulence genes and hypermucoidy, found only 13% of rmp+ strains were actually hypermucoid (mostly due to mutation of the rmpA gene), these were also virulent in a neutrophil assay although unfortunately they do not report clinical severity in patients **We now cite this reference in the Introduction and Discussion.**

iv) “Genomic epidemiology study of *Klebsiella pneumoniae* causing bloodstream infections in China” (doi: 10.1002/ctm2.624), report clinical results for n=239 blood stream infections with Kp including 129 with hv genes of which 110 were hv in a Galleria model; hv strains were associated with increased CRP, neutrophil counts, total white blood cells; other clinical variables showed potential associations also **This study has now been cited in the Introduction.**

Line 120 - why define the term “highly antibiotic resistant” that is based on resistance to a single drug class? Why not just refer to these directly as third-generation cephalosporin resistant (3GCR), and avoid the need to define a term that obscures what is meant? This would be much clearer for readers. **The text has been updated as suggested.**

Line 146 - the abbreviation “CPE” is typically used to delineate carbapenemase-producing Enterobacteriaceae, I find it confusing to redefine this as ‘carbapenemase’; why not just use the word ‘carbapenemase’? **CPE was removed and this paragraph has been updated for clarity.**

Lines 144-149 - Again it is not clear to me what is meant by ‘MDR Kpn’. In the intro it was stated that MDR Kpn carry ESBLs; but here it is stated that 90% carry resistance determinants for ≥ 3 drug classes; 78.6% carried an ESBL, others had a carbapenemase but some lacked both ESBL and carbapenemase... so what is the definition of MDR here? Clearly it is neither ‘ ≥ 3 drug classes’ nor presence of ESBL or carbapenemase, so how is it defined? **We thank the reviewer for pointing this out. We now define MDR Kpn as clinically relevant phenotypic resistance to ≥ 3 drug classes. We have revised the text throughout the manuscript to ensure that we use “MDR” in a manner consistent with this definition. Note that in the indicated paragraph we are describing the resistance genes these strains contain (as opposed to resistance phenotypes).**

Line 150 - typo, ‘atleast’ should be ‘at least’ **The text has been updated as suggested.**

Lines 308-309 state that OC1092 had a frameshift in *wzi* and was presumably less virulent for that reason, please clarify was it confirmed whether this strain produces capsule? This strain did not produce capsule as measured by uronic acid production (Figure 5B & Table S2). Phenotypic data for individual strains has been added to Table S2.

Methods: please add version numbers for all software Version numbers for all software have been added to the methods.

Methods: The 'Mouse studies' section lacks detail on how lethal dose was determined (and also lacks critical details on bacterial passage, as noted above) We have now revised this section to add more detail. LD50s values were determined using the *drc* package in R. Additional details on mouse strains, dosing, and deaths are found in Table S5. A description of bacterial passaging has been added to the Methods section, as described above.

Paragraph beginning line 235 - only one of the convergent isolates was hmv; this is one of three isolates with a *rmpADC* locus noted in Table 2. Was the locus intact in the other 2 strains (ST23 DHQP1701672 and ST16 KPN226)? (The other 9 isolates would not be expected to be hmv as they lack the *rmp* locus.) Was this checked in the culture from which DNA was extracted? We have now examined the *rmpADC* locus of KPN226 and identified an early frame shift mutation in *rmpA*, which likely disrupts its function. We also examined DHQP1701672 and identified two SNVs 200 bp upstream of *rmpA* and one SNV in the intergenic region between *rmpA* and *rmpD*. This information has been added to the manuscript. Unfortunately, hmv was not determined on the cultures used for sequencing, but we have not noted variations in hmv in all our subsequent subcultures from the original freezer vial from which sequencing was performed. For this reason, we feel it would be unlikely that the genome sequence of these isolates would differ from the sequences of the isolates on which subsequent hmv studies were performed.

Reviewer #2:

1. It is surprising to the reviewer that even with having the *rmp* locus the convergent strains are hmv negative. Is the *rmp* locus being expressed? Comparison could be to an hvKp isolate. We have not measured the expression of the *rmp* operon in these isolates but have now looked at their sequences and noted possible explanations for their hmv phenotypes. See the preceding paragraph for details.

Furthermore, did the authors determine if there were any mutations in the capsule locus, specifically in *wzc*, to which RmpD was recently shown to interact (Ovchinnikova 2023)? This would also provide insight in to the hmv negative phenotype. We thank the reviewer for this suggestion. We have now examined the *wzc* gene of DHQP1701672, but no mutations were noted.

Also, in their uronic acid assay, are the samples adjusted to OD600? As absorbance can be affected by capsule levels, adjustment to CFU count should be considered. We appreciate this suggestion. Yes, the samples were adjusted by OD600 and not CFU. We considered adjusting to CFU initially, but based on our experience with this assay we think OD600 is acceptable for several reasons. The capsule assay performed was slightly modified from Mike et al (PLoS Pathogens 2021). The only modification was that we adjusted by OD600 prior to extracting capsule rather than normalizing by OD600 after capsule extraction. We found that adjusting by OD600 prior to extraction was more accurate than normalizing mathematically after extraction. We think this is due to the nature of the capsule extraction procedure; differences in the amount of biomass included in the assay can affect the efficiency of capsule extraction. Since it is not possible to normalize by CFU prior to extraction but only after extraction, we decided to adjust by OD600 prior to capsule extraction. In addition, we found that there is very little strain-to-strain variation of OD600 vs CFU for the strains included in this manuscript. For example, for each isolate included in this manuscript, we were able to use the same OD to CFU curve to calculate our dosing for our animal experiments with good accuracy (as validated by plating of the inoculum). Using this curve, our dosing was accurate even for OC1092, which is a nonencapsulated strain. For these reasons, we think adjusting to OD600 prior to extraction is acceptable.

2. The string test performed is highly qualitative. As it is only 12 strains that were further selected for testing, the authors should perform the spin down hmv assay as described by Walker et al 2020, which is a more stringent and quantitative assay for hmv phenotype. We have now performed quantitative centrifugation testing on all 40 strains included in this study and added those results to Figure 5, as suggested. Methods for quantitative centrifugation were also updated.

3. It is not clear to the reviewer if they observed any differences in their serum killing between different isolates? Which is surprising as Pomakova et al 2011 and other studies have shown that hvKP isolates have higher resistance. The reviewer is correct that it has been previously described that hvKP isolates are more resistant to serum killing than cKP isolates. However, most of those studies only include traditional hvKP isolates such as K1-ST23 or K2-ST86 hvKP and make comparisons with cKP isolates of unknown origin. Our study includes an intentionally diverse cohort of hvKP and cKP clinical isolates. Indeed, our hvKP isolates are highly serum resistant and many of our cKP isolates were relatively serum sensitive (Fig. 5A). However, many cKP isolates were also highly serum resistant. We feel that the variability in serum resistance among cKP isolates may account for the different conclusions of our study and other published studies.

For all the panels in Figure 5, the reviewer would suggest that they show individual strains as a bar graph, as it is important to see if there is a correlation between any of the tested strains phenotypes and their genotypes. (An example would be Russo et al 2021) By pooling everything it dilutes the results. They can still show the pooled data as well. Unfortunately, the large number of isolates in Figure 5 make a graph with individual bars somewhat messy and difficult to interpret. We have therefore placed the

corresponding data in a table (Table S2), which will allow the reader to compare the genotypes vs. phenotypes of individual isolates.

4. A concern to the reviewer is in regards to strain(s) site of isolation. It appears that ~50% of the isolates were from urine compared to other sites of infection. This could bias their findings. This should be addressed in the discussion section. It would also be useful to list the site of isolation for the tested strains, rather than have the reader dig through supplemental data to figure out the site of isolation for the 12 tested strains. We apologize for making this information difficult to access. **We have now added the site of isolation to Table 2. Only four of the 12 convergent isolates were from urine. The others were from the bloodstream (4), lung (3), or bile (1). In total, these isolates represent infections at a variety of body sites.**

Minor point.

a. In line 148 should that also include carbapenems? **The text was updated to include carbapenems.**

REVIEWERS' COMMENTS

Reviewer #1 (Remarks to the Author):

The authors have addressed all comments raised.